# Imp interacts with Lin28 to regulate adult stem cell proliferation in the *Drosophila* intestine

**Perinthottathil Sreejith**[1,2], **Sumira Malik**[2¤], **Changsoo Kim**[2], **Benoît Biteau**[1] *

**1** Department of Biomedical Genetics, University of Rochester Medical Center, Rochester, New York, United States of America, **2** School of Biological Sciences and technology, Chonnam National university, Gwangju, South Korea

¤ Current address: Amity Institute of Biotechnology, Amity University Jharkhand, Ranchi, India
* benoit_biteau@urmc.rochester.edu

**Data Availability Statement:** All relevant data are within the manuscript and its Supporting Information files.

**Funding:** This work was supported by the grant R01GM108712 from the National Institutes of

## Abstract

Stem cells are essential for the development and long-term maintenance of tissues and organisms. Preserving tissue homeostasis requires exquisite control of all aspects of stem cell function: cell potency, proliferation, fate decision and differentiation. RNA binding proteins (RBPs) are essential components of the regulatory network that control gene expression in stem cells to maintain self-renewal and long-term homeostasis in adult tissues. While the function of many RBPs may have been characterized in various stem cell populations, how these interact and are organized in genetic networks remains largely elusive. In this report, we show that the conserved RNA binding protein IGF2 mRNA binding protein (Imp) is expressed in intestinal stem cells (ISCs) and progenitors in the adult *Drosophila* midgut. We demonstrate that Imp is required cell autonomously to maintain stem cell proliferative activity under normal epithelial turnover and in response to tissue damage. Mechanistically, we show that Imp cooperates and directly interacts with Lin28, another highly conserved RBP, to regulate ISC proliferation. We found that both proteins bind to and control the InR mRNA, a critical regulator of ISC self-renewal. Altogether, our data suggests that Imp and Lin28 are part of a larger gene regulatory network controlling gene expression in ISCs and required to maintain epithelial homeostasis.

## Author summary

Stem cells are essential to maintain healthy organs. However, dysregulation of their function is a potential major driver of diseases, including cancer and neurodegeneration, and significantly contributes to the aging process. For these reasons, numerous mechanisms control the ability of stem cells to divide and give rise to functional daughter cells. In this study, we used the *Drosophila* fruitfly as a genetically amenable experimental model to characterize the function of a conserved protein, the IGF2 mRNA binding protein, in the regulation of adult intestinal stem cells. We found that it is essential for stem cell proliferation under normal conditions and in response to tissue damage. We also report that it

Health https://www.nigms.nih.gov/ to BB and the grant NRF-2021R1A2C1010334 from the National Research Foundation of Korea https://www.nrf.re. kr/eng/main/ to CK. The funders had no role in study design, data collection and analysis, decision to publish, or preparation of the manuscript.

**Competing interests:** The authors have declared that no competing interests exist.

interacts with another known regulator, Lin28. Importantly, these two factors largely control stem cell biology and development in mammals, including humans, and are often dysregulated in cancer. This suggests that our work is shedding new light on the conserved mechanisms that maintain long-term stem cell function across organisms.

## Introduction

Self-renewal is essential for normal and pathological stem cell function and relies on complex gene regulatory networks. Thus, understanding the molecular mechanisms of gene regulation in stem and progenitor cells is critical for designing strategies to reprogram somatic cells, promote tissue regeneration, treat degenerative diseases, or understand the tumorigenesis process. RNA binding proteins (RBPs) have emerged as key regulators of cell pluripotency and self-renewal [1, 2]. Particularly, IGF2 mRNA binding proteins (IGF2BPs / IMPs) and Lin28 are two families of RBPs that have been identified as critical regulators of gene regulation in stem cells and progenitors, controlling many essential aspects of their biology during development and in adult tissues [3, 4]. On one hand, IMPs are a group of highly conserved RBPs that regulate mRNA targets by promoting their stability either by preventing mRNA degradation, by localizing mRNA to cytoplasmic granules, or by blocking siRNA/miRNA mediated silencing [3]. As a result, IMPs are essential to maintain stem cell identity and function in many biological contexts [3, 5]. On the other hand, Lin28, a highly conserved RBP, was identified early as one of the critical factors sufficient to reprogram human somatic cells into pluripotent stem cells and regulates metabolic genes in these cells [6, 7]. Mechanistically, Lin28 are best characterized for its regulation of microRNA let-7 biogenesis but, more recently, Lin28s have been shown to associate with mRNA-ribonucleoprotein (mRNP) complexes and acts as regulators of mRNA stability and translation by binding to many transcripts in embryonic stem cells (ESCs) and other stem cells [8, 9].

RBPs, including IMPs and Lin28s, have been extensively studied in developing tissues, adult stem cells or in tumor cells. However, beside the individual function of RBPs, increasing evidence suggest that the interaction between RBPs is critical in defining the fate of RNA and gene expression [10, 11]. Investigating how RBPs cooperate and/or compete to regulate the post-transcriptional gene network thus becomes crucial to understand the establishment and maintenance of different cellular phenotypes, like stem cell pluripotency and self-renewal. Recently, several studies have identified genetic and biochemical interactions between IMPs and Lin28s in various stem and progenitor populations [12–14]. However, many questions remain regarding the role of this potential interaction in adult tissue stem cells, or in cancer cells where higher expression of these proteins have been associated with disease state [15–21].

Our work takes advantage of *Drosophila* as a genetic model, where both Imp and Lin28 proteins are highly conserved. While there are three IMP (IMP1, 2, 3) orthologues, and two Lin28 (Lin28A, B) orthologues in mammals, the fly genome encodes only one Imp and one Lin28 gene. In recent studies, we and others identified the role of both these RBPs in the regulation of several adult fly stem cell populations. In the testis, Lin28 is expressed in the niche cells and is required for the self-renewal of testis germline stem cells indirectly via regulating self-renewal factor Unpaired (upd), independently of the microRNA let-7 [22]. In the same cells, Imp also binds to and control the stability of the Upd messenger [23], suggesting that Imp and Lin28 may cooperate to control the function of the stem cell niche. To investigate the function of Imp in stem cells themselves, we turned to adult intestinal stem cells (ISC). The adult midgut is maintained by multipotent ISCs that can give rise to progenitors committed to the absorptive enterocyte (ECs) fate, the Enteroblasts (EBs), as well as progenitors committed to the

secretory enteroendocrine cell fate (pre-EEs and EEs). In the intestinal epithelium, Lin28 is highly enriched in intestinal stem cells and is required for adult stem cell expansion by regulating the mRNA encoding for the insulin receptor (InR) [24]. However, while the expression of IMP in these cells has been suggested [25], its function in ISCs remain to be thoroughly investigated.

In addition to IMP and Lin28, several post-transcriptional regulators of gene expression controlling intestinal tissue homeostasis have been identified in ISCs. For example, while Lin28 drives symmetric cell division by enhanced insulin signaling, independently of let-7, FMRP (Fragile X Mental Retardation protein), another RBP, acts to oppose Lin28-driven proliferation [24, 26]. Similarly, Tis11, an Adenine-Uridine Riche Element (ARE) binding protein promote RNA destabilization to restore proliferative homeostasis after tissue repair [27]. Lastly, the role of microRNAs in the ISC lineage is starting to be explored; for example, expression of miR-8 and miR-305 controls cell differentiation and self-renewal respectively by affecting the expression of specific components of the conserved signaling pathways that regulate these processes [28, 29]. Altogether, these observations suggest that post-transcriptional regulators of genes expression form a regulatory network that largely remains to be explored in ISCs.

Here, we report that Imp is expressed in *Drosophila* adult intestinal progenitors, and that it is required and sufficient cell-autonomously to promote proliferation of ISCs. We found that Imp and Lin28 genetically cooperate to control ISC proliferation and that these two proteins can physically interact. We propose a model in which Imp and Lin28 act non-redundantly to regulate the stability of the InR mRNA in to control adult intestinal proliferative homeostasis under normal conditions and in response to tissue damage.

## Results

### Imp expression is highly enriched in adult intestinal progenitors

Based on the functional interaction between IMPs and LIN28s in many stem cell populations, on the function of both Imp and Lin28 in the fly testis niche, on the expression of Lin28 in ISCs, and a recent report [25], we hypothesized that Imp could play a role in controlling ISC function. First, to investigate the expression pattern of Imp in the adult intestinal epithelium, young adult midguts were stained with an antibody directed against the endogenous Imp protein. We found significant Imp expression in all progenitor cells (ISC and EBs), as well as weak expression in EEs (Fig 1). This was first demonstrated by colocalization of Imp protein with Delta, a well-established ISC marker, as well as low signal in prospero-positive EEs (Fig 1A). Also, we show that Imp is expressed in all escargot-positive cells, using the esg-Gal4>UAS-GFP;tubulin-Gal80ts (esgGFP[ts]) driver line (Fig 1B). We knocked-down the expression of Imp in esg-positive cells, using the temperature-inducible esgGFP[ts] driver and a dsRNA construct directed against Imp (UAS-Imp[RNAi]); the loss of Imp signal in this condition confirmed the specificity of our staining and the efficacy of the RNAi line (Fig 1B). To support the anti-Imp antibody staining, we next used the protein trap line GFP-Imp [30]. We found the expression of the Imp fusion protein in all intestinal progenitors, as shown by the co-expression with the ISC- and EB- specific esg-LacZ reporter (Fig 1C) and colocalization with the anti-Imp staining (Fig 1D). Altogether these data indicate that Imp is enriched in ISCs and EBs in the midgut epithelium, suggesting it may play an essential regulatory function in ISCs.

### Imp expression is required and sufficient cell-autonomously for ISC proliferation

Based on its expression in ISCs and EBs, we asked whether, like Lin28, Imp is required for ISC proliferation. Since Imp mutants are homozygous lethal [31], we used MARCM (Mosaic

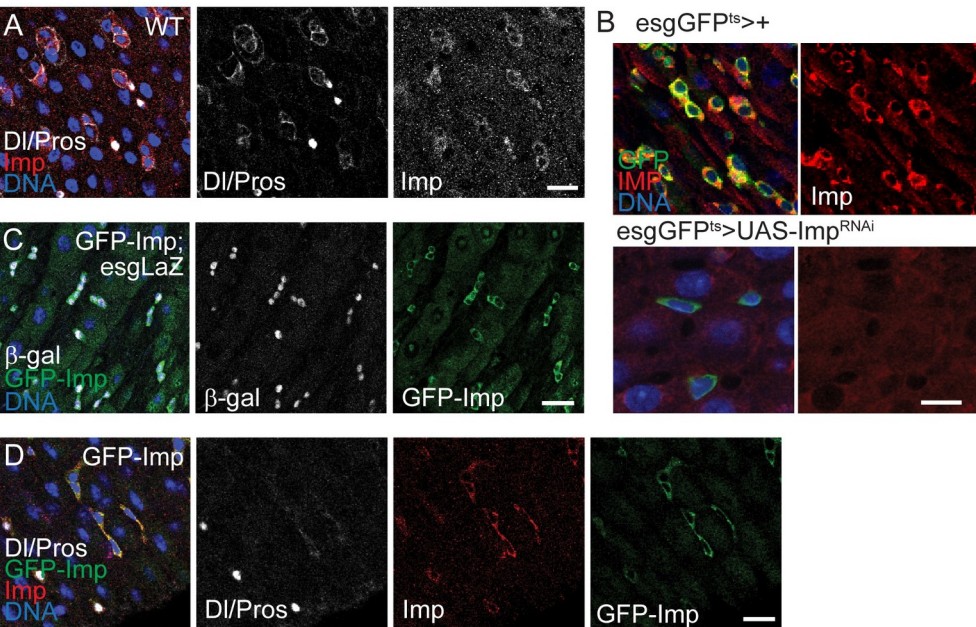

**Fig 1. Imp expression in adult intestinal stem cell progenitors.** (A) Confocal images of 7-day old wild type intestinal epithelium stained with antibodies specific to Imp and Delta (Dl, ISC marker) and Prospero (Pros, enteroendocrine cell marker). (B) Confocal images of 5-day old adult guts (esg-GFP$^{ts}$) stained with antibody specific to Imp shows the expression of Imp in Stem cell progenitors where GFP is expressed in ISCs and EBs. Adult specific knockdown of Imp using Imp RNAi (esg-GFP$^{ts}$>ImpRNAi) leads to loss of Imp staining in the stem cell progenitors compared to controls (esgGFP$^{ts}$>+). (C) Confocal images of the intestine of 7-day old trans-heterozygote for GFP-Imp-Trap and progenitor specific Lac-Z reporter (GFP-Imp;esg-LacZ) stained with antibody specific to GFP and β-galactosidase show the expression of Imp in all LacZ-positive ISCs and EBs. (D) Confocal images of 7-day old GFP-Imp gut co-stained for delta, Prospero, Imp and GFP antibody confirms the expression of Imp in ISCs and EBs. In all panels, scale bar: 10μm.

analysis with a repressible cell marker; [32]) clonal analysis to induce and trace *Imp⁷* homozygous mutant ISC lineages in heterozygous adults. Intestines were analyzed at different time points after clonal induction (days after heat shock) to assess the number of labelled cells per clones. In control intestines, we observed a persistent increase in the size of marked clones reporting the proliferative activity of individual ISCs (Fig 2A). However, *Imp⁷* homozygous mutant clones remained constantly small, mostly composed of 1 to 2 Dl-positive Sox21a-positive ISCs and very few Sox21a EBs [33], even 30 days after clonal induction (Fig 2A). This demonstrates that Imp is required cell autonomously for ISC proliferation but dispensable for their survival under normal conditions.

While the intestinal epithelium remains relatively quiescent at homeostasis, ISC proliferation can be increased in response to various external stimuli, such as exposure to toxicants or bacterial infection [34–36]. The chemical stressor dextran sulfate sodium (DSS) induces a rapid and robust ISC response [34]. Therefore, we tested the role of Imp in intestinal progenitors in this stress paradigm. We combined UAS-Imp$^{RNAi}$ with cell type specific drivers to conditionally knockdown Imp in ISCs and/or EBs and ISC proliferation by feeding these transgene-expressing flies 4% DSS or sucrose (control). Proliferation in the intestinal epithelium of these animals was measured by counting the number of cells positive for mitotic marker phospho-histoneH3 (pH3). We found that loss of Imp specifically in ISCs, not in EBs, lead to blockage of proliferation: DSS-induced proliferation is inhibited when Imp$^{RNAi}$ is driven by the esgGFP$^{ts}$ (ISCs+EBs; esgGal4>UAS-GFP,tub-Gal80$^{ts}$) or the ISC-YFP$^{ts}$ (ISCs only; esGal4>UAS-YFP,tub-Gal80$^{ts}$,GBE-Su(H)Gal80), but not when we used the GBE-GFP$^{ts}$

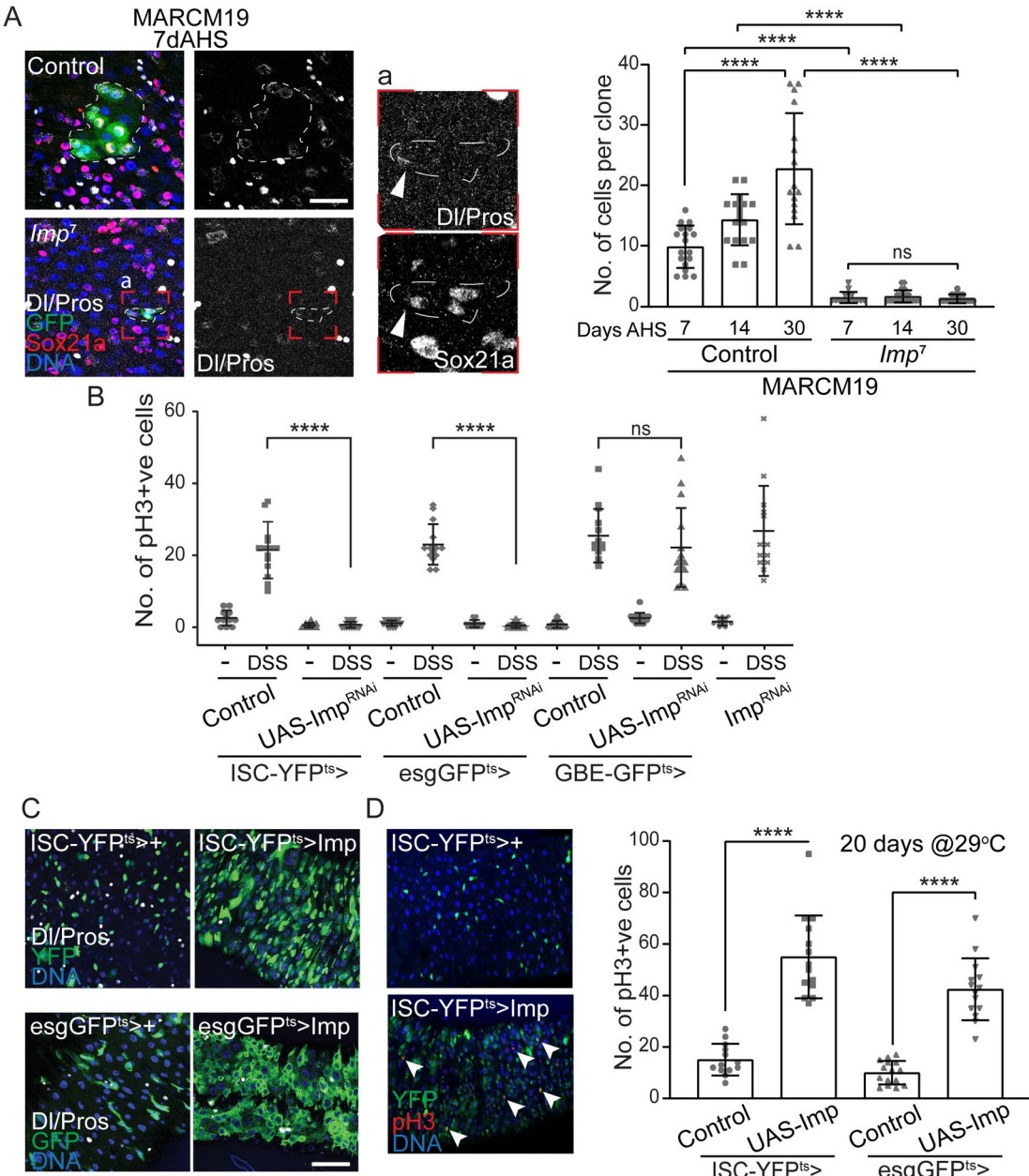

**Fig 2. Imp is required cell autonomously in Intestinal Stem cells.** (A) MARCM clonal analysis of control and Imp homozygous null ISCs shows that Imp is essential for ISC proliferation. Clones are labeled by GFP expression (green), Sox21a identifies ISCs and EBs, Delta (Dl) specifically stains ISCs and Prospero (Pros) highlights enteroendocrine cells. Insert 'a' presents a higher magnification image of an *Imp* mutant clone, illustrating the presence Dl-positive and Sox21a-positive cells in these very small GFP+ cell clusters. Box plot represents the distribution of number of cells per clone in an age-dependent manner. Imp mutant clones fail to grow, even after 30 days of clonal induction (days after heat shock, AHS). Student's t-test **** p-value = < 0.0001; ns = 0.426. (B) Imp is required specifically in ISCs for cell proliferation. Cell specific knockdown of Imp in ISCs, but not in EBs, abolishes DSS- induced cell division. Proliferation is measured by the number of phospho-Histone H3 (pH3) positive cells per gut 48-hour exposure to DSS, in control animals or when Imp is knocked-down in ISCs (ISC-YFP^ts), ISCs+EBs (esgGFP^ts) or EBs (GBE-GFP^ts). (C, D) Over-expression of Imp in ISCs (ISC-YFP^ts) or ISCs+EBs (esgGFP^ts) is sufficient to promote cell proliferation, as shown by the expansion of the esg-positive and Dl-positive cells (C) and the increased number in pH3-positive mitotic cells (arrowheads, D). Delta (Dl) specifically stains for ISCs and Prospero (Pros) stains for EEs. DNA is stained by Hoechst. In B and D, each data point represents the number of pH3-positive cells in a gut; n>10 guts per genetic and treatment conditions. Student's t-test **** p-value = < 0.0001; * p = <0.05; ns p = 0.3562. In panels A and C, scale bar: 10μm.

driver (EBs only; GBE-Su(H)Gal4>UAS-GFP,tub-Gal80^ts) (Figs 2B and S1A). Next, we exposed esgGFP^ts>UAS-Imp^RNAi to the ROS-producing compound paraquat (PQ), to promote stem cell division [37], and observed a similar block of ISC proliferation in this oxidative stress paradigm (S1B Fig). This confirmed that, like under homeostatic conditions, Imp is required cell-autonomously in ISCs to promote cell division after tissue damage. This supports the notion that it acts downstream of the stress-responsive transcription factor ets21c in these cells [25].

Conversely, we found that long-term over-expression of Imp using the esgGFP^ts or ISC-YFP^ts drivers results in very high numbers of mitotic cells in the intestine, demonstrating that Imp is sufficient to promote ISC proliferation cell-autonomously (Fig 2C).

Altogether, our data establish that Imp is a critical regulator of ISC proliferation during homeostatic tissue turnover and in response to tissue challenge.

## Imp and Lin28 cooperate to control ISC proliferation

In the testis, Imp and Lin28 are part of a regulatory network that control the expression of the self-renewal factor Upd in hub cells. Thus, we genetically interrogated the relationship between Imp and Lin28 in ISCs.

First, we asked whether increased levels of Imp and Lin28 can cooperate to promote ISC proliferation. As opposed to Imp long-term over-expression (Fig 2C), two-day overexpression of each factor individually, using the esgGFP^ts driver, results in a significant but limited increase in proliferation (Fig 3A). However, when combined, the co-overexpression of Imp and Lin28 synergistically increases ISC proliferation, as seen by high levels of pH3+ cells, 2 days after transgenes induction, and the large expansion in the number of esg>GFP+ cells, 5 days after induction. This results in flies with increased sensitivity to DSS treatment, as shown by their significantly reduced survival under these culture conditions (S2A Fig).

Next, we asked whether the combined loss-of-function of Imp and Lin28 may affect ISC and tissue homeostasis. We used the esgGFP^ts driver to express UAS-Imp^RNAi+-UAS-Lin28^RNAi in ISC/EBs starting in young adult flies. Like Imp^RNAi expression alone, this manipulation completely blocks DSS-induced ISC proliferation (S1A Fig). However, this combination also significantly decreases the number of esg-positive cells in the intestine, compared to wild-type animals or flies expressing only one of the RNAi constructs (Fig 3B). In addition, the combined knock-down of Imp and Lin28 results in dramatically shorter-lived flies, with median lifespan of 8 to 10 days at 29°C (Fig 3C). Of note, the combined knock-down does not affect the sensitivity to DSS treatment (S2A Fig), suggesting that ISC proliferation is required for long-term maintenance but dispensable for stress response.

These data suggest that Imp and Lin28 cooperate to maintain intestinal homeostasis. Thus, we next tested whether they may act redundantly in ISCs. To this end, we performed genetic rescue experiments using MARCM clones (S2B Fig). As expected, Imp expression can significantly rescue the proliferation of *Imp*^7 homozygous clones and Lin28 expression restores *Lin28*^Δ1 clone growth. However, when Lin28 was expressed in *Imp* null mutant clones or, conversely, when Imp was expressed in *Lin28* null clones, only minimal rescue was observed (Fig 3B), suggesting that Imp and Lin28 cannot optimally compensate for the absence of the other factor.

Of note, we observed that over-expression of Imp using the esgGFP^ts driver leads to a detectable increase in Lin28 in intestinal progenitors and, conversely, that over-expressing Lin28 results in increased Imp protein expression (S3 Fig). This suggests that these two RBPs regulate each other directly or indirectly.

Together, these observations lead us to conclude that Imp and Lin28 are not redundant, but rather cooperate to control ISC proliferation and maintenance.

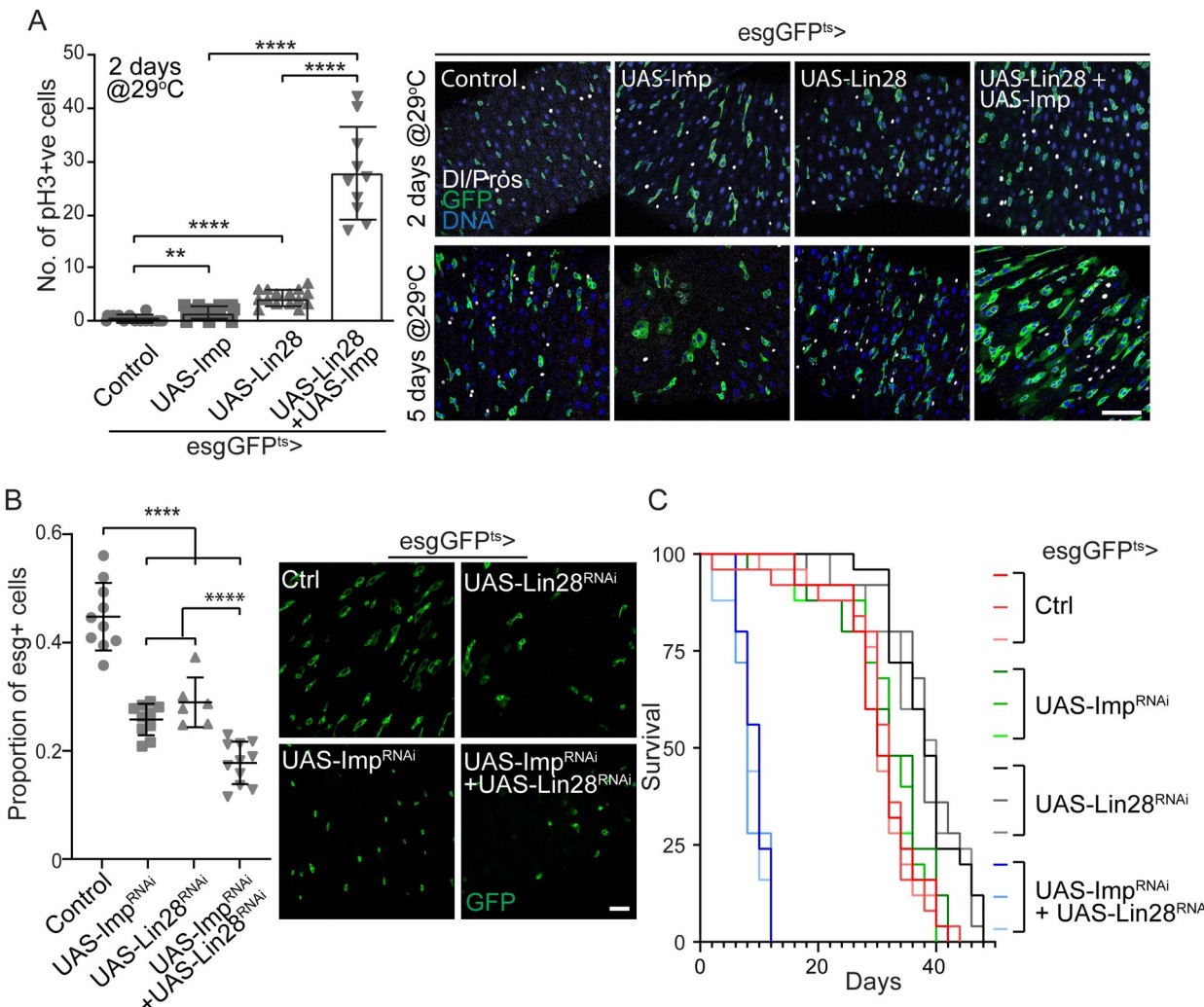

**Fig 3. Imp and Lin28 cooperate to regulate ISC proliferation.** (A) Co-expression of Lin28 and IMP leads to synergistically increased ISC proliferation. Box plot representing the quantification of the number of pH3 positive cells per gut that shows that ISCs/EBs specific co-expression of Imp and Lin28 leads to high number of mitotic cells 2 days after transgene induction. Representative images are shown to illustrate the expansion of esg-positive cells 5 days post induction of both transgenes. GFP is expressed in ISCs and EBs, Delta stains for ISCs and Prospero (Pros) stains for EEs. DNA is stained by Hoechst. Student's t-test **** p-value = <0.0001; ** p = 0.0036. (B) Double knockdown of Imp and Lin28 results in a long-term loss of progenitors in adult guts, 7 days after transgene activation. Plot presenting the comparison of the percentage of esg-positive in the posterior midgut (measured by the ratio of GFP positive cells vs the total number of cells, counted by DNA-positive nuclei) showing a reduced number of progenitors when either Imp or lin28 are knocked down, which is exacerbated when both Imp and Lin28 are knocked down. Representative confocal images are shown to illustrate the changes in the number of GFP-positive cells. (C) Combined loss of Imp and Lin28 reduces adult female lifespan. Adult specific knockdown of Imp and Lin28 in esg-positive cells leads to accelerated death when flies are reared at 29˚C. Kaplan-Meier survival curves of three populations of 25 flies are shown for each genetic condition. In A and B, scale bar: 10μm.

## Imp regulates InR expression in ISCs

In hub cells, Imp and Lin28 both bind to the upd mRNA and regulate its stability [22]. In ISCs, Lin28 controls cell proliferation, at least in part, by regulating the InR messenger [24]. We then hypothesized that, similarly, Imp regulates InR and that the defects in ISC proliferation observed in Imp loss-of-functions are caused by reduced insulin/IGF-1 signaling. To first test this model, we asked whether increased of InR expression in *Imp* mutant clones could rescue the strong proliferation defect that we observed (Fig 2A). Using the MARCM technique, we drove the expression of the wild-type InR protein (UAS-InR) or an activated form of the

receptor (UAS- InR[act]) in *Imp*[7] homozygous mutant clones (Fig 4A). We found that expression of either of these InR transgenes is sufficient to fully rescue ISC proliferation, as shown by the significant clone growth observed 7 and 14 days after induction.

Previously it was shown that Lin28 can physically interact with the 3'UTR of the InR mRNA to regulate its stability [24]. Thus, we reasoned that, like Lin28, Imp may also affect the steady state level of the InR transcript. We performed qRT-PCR in the same conditions where we observed a synergistic effect of Imp and Lin28 co-expression on ISC proliferation (Fig 3A). No significant change in InR expression is observed when Imp or Lin28 are over-expressed individually using the esgGFP[ts] driver. However, a two-day combined induction of the UAS-Imp+UAS-Lin28 transgenes, which does not cause a significant change in the number of esg+ cells (Fig 3A), results in a marked increase in the level of InR mRNA in total intestinal RNA extracts (Fig 4B). This strongly suggests that the levels of InR mRNA are synergistically induced by Imp and Lin28 in ISCs and EBs. Using immunostaining, we confirmed the ability of Imp or Lin28 over-expression to individually cause the accumulation of significantly higher InR protein levels in esg-positive cells, 7 days after transgene induction (S4A and S4B Fig). Conversely, we found that, like Lin28 null animals, *Imp*[7] heterozygotes show reduced levels of *InR* transcripts in the intestine compared to wild-type controls (Fig 4C) and lower InR protein expression in Dl-positive cells (Fig 4D). Finally, we tested the binding of Imp to the InR mRNA in the adult gut. We performed RNA-Immunoprecipitation followed by qRT-PCR in intestinal extracts from animals expressing either the GFP-Imp or Lin28-Venus protein fusions under their respective endogenous promoters. We found a significant enrichment of *InR* transcripts in the anti-GFP (GFP-Imp extracts), or anti-Venus (Lin28-Venus extracts) immuno-precipitates compared to control non-specific antibodies (Fig 4E), strongly suggesting that Imp and Lin28 can both bind the InR mRNA in intestinal progenitors.

Altogether, these data strongly support the notion that Imp binds to the InR mRNA to promote its stability and allow ISC proliferation.

## Lin28 and Imp can directly interact

We found that Imp and Lin28 are both expressed in intestinal progenitors, are essential for ISC proliferation and regulate InR mRNA levels. This raised the possibility that the Imp and Lin28 proteins directly interact to control expression of their targets, including InR. Interestingly, previous studies reported the direct interaction between IMP3 and LIN28B in hematopoietic stem and progenitor cells (HSPCs), and IMP1 and LIN28A in neuronal progenitors [13, 14]. To demonstrate the ability of *Drosophila* Imp and Lin28 proteins to interact, we used several distinct experimental models. Firstly, we carried out a yeast two hybrid assay and found that Lin28 and Imp protein fusions can form a stable complex in yeast. Co-expression of Imp-TAD+Lin28-DBD or Imp-DBD+Lin28-TAD (TAD: Transcriptional Activation Domain of the transcription factor GAL4; DBD: LexA DNA Binding Domain) is sufficient to promote the expression of a LexAop-driven beta-Galactosidase reporter in S. cerevisiae cells (Fig 5A). Next, we performed co-immunoprecipitation in cultured *Drosophila* S2 cells. We found that, when Flag-tagged Imp and HA-tagged Lin28 proteins are co-expressed, both proteins can be co-precipitated using either an anti-Flag antibody or an anti-HA antibody (S5 Fig). Importantly, when the protein extract is pre-treated with RNAse, the co-precipitation is not affected suggesting the Imp-Lin28 protein interaction is RNA-independent in fly cells (Fig 5B). Finally, we confirmed the ability of Imp and Lin28 proteins to interact by GFP fragment complementation assay in HEK293T cells. Imp and Lin28 fusion proteins with the C-terminal or N-terminal domains of GFP were transfected in cells. Only when the combinations GFP-ImpNter+Lin28-GFPCter or GFP-ImpCter+Lin28-GFPNter are co-expressed, can

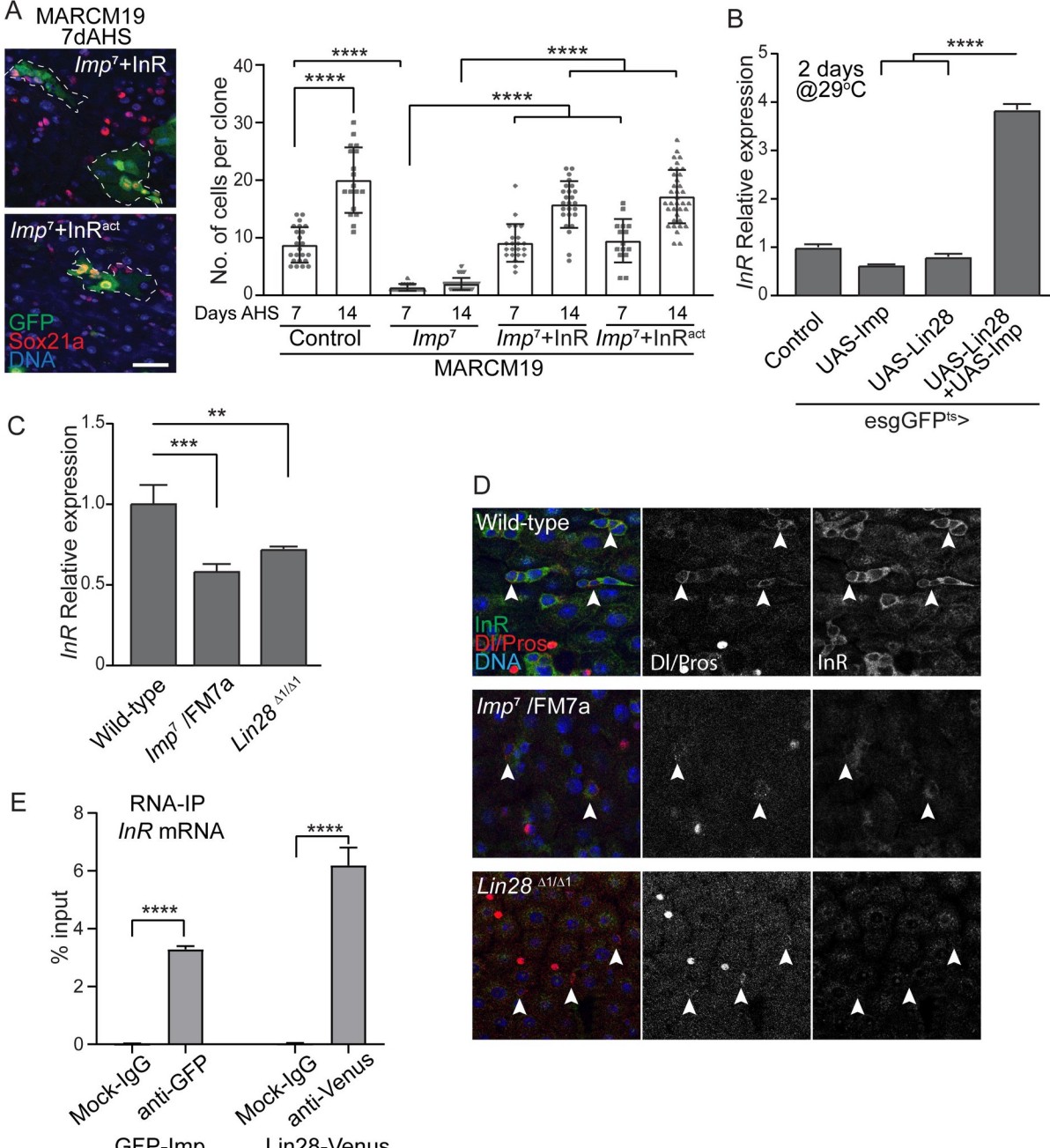

**Fig 4. Imp regulates InR mRNA expression in the intestinal epithelium.** (A) Confocal images of MARCM clones illustrating the growth of Imp7 homozygous mutant clones expressing wild-type and constitutive active forms of the insulin receptor (InR). GFP expression labels ISC clones, Sox21a stains for the progenitors and DNA is stained with Hoechst. The number of cells per clones in the different genetic backgrounds, 7 and 14 days after heat shock induction (AHS), are shown, demonstrating that activating InR signaling is sufficient to restore ISC proliferation in Imp mutants. Student's t-test **** p-value = <0.0001 Scale bar: 10μm. (B) Relative expression of the InR mRNA, normalized to escargot, in the 2-day old gut from control animals and when Imp and/or Lin28 are expressed in esg-positive cells. Imp+Lin28 co-expression is sufficient to drive the expression of InR, suggesting that both proteins regulate InR synergistically. Student's t-test **** p-value = <0.0001. (C) qPCR quantification of the InR mRNA, normalized to rp49, in 7-day old gut RNA extracts shows reduced InR transcript levels in Imp heterozygotes and Lin28 homozygous mutants. Student's t-test *** p-value = 0.0005, ** p = 0.0036. (D) Representative confocal image showing the decreased levels of InR protein detected by immunostaining (green) in intestinal progenitors of *Imp* and *Lin28* mutants, compared to wild-type animals. Dl-positive ISCs are indicated by arrowheads. DNA is stained with Hoechst. (E) Imp and Lin28 bind to InR transcripts in adult guts. 7-day old guts were subjected to RNA-immunoprecipitation using either GFP-Imp or Lin28-venus, followed by qPCR. Significant levels of InR mRNA, normalized to input materials, are pulled down using both proteins, compared to controls. Student's t-test **** p-value = <0.0001.

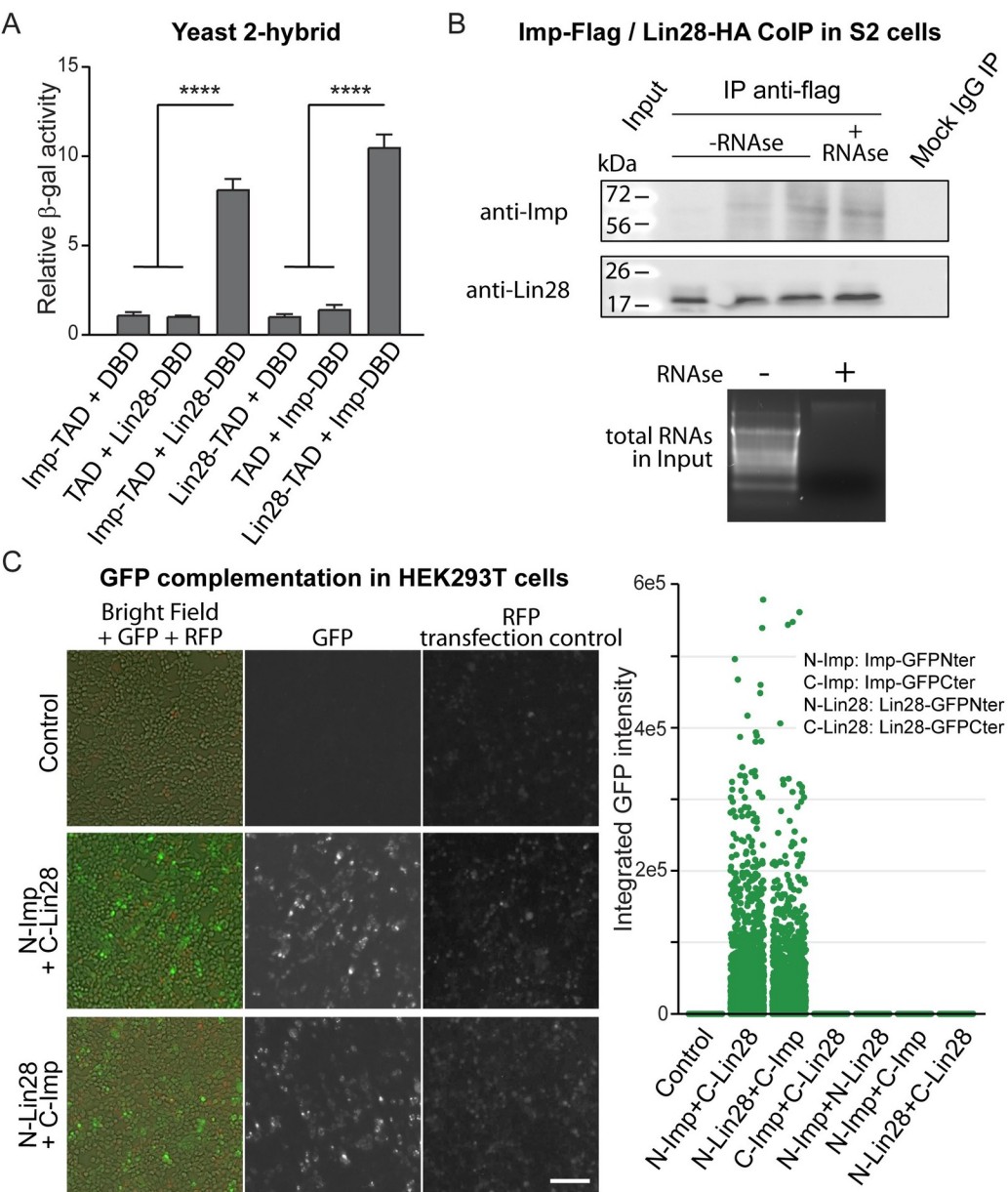

**Fig 5. *Drosophila* Imp and Lin28 physically interact.** (A) Imp and Lin28 can form a stable complex in yeast cells, as shown by 2-hybrid assay. Quantification of the β-galactosidase activity when Imp and Lin28 protein fusion with the Gal4 Transcriptional Activation Domain (TAD) or the LexA DNA Binding Domain (DBD). Significant reporter activity is detected only when the Imp-TAD+Lin28-DBD or Imp-DBD+Lin28-TAD protein combinations are expressed. Student's t-test **** p-value = <0.0001. (B) Co-immunoprecipitation of HA-tagged Lin28 and FLAG-tagged Imp proteins expressed in cultured *Drosophila* S2 cells. Anti-Flag precipitation pulls-down both protein fusions. RNase treatment of the lysate does not affect the stability of the complex. Total RNAs stained with ethidium bromide demonstrate the efficacy of the RNase treatment. (C) Fluorescence images of GFP complementation assay in HEK293T cells show that only when the appropriate combinations of Lin28 and Imp protein fusions (GFP-ImpNter+Lin28-GFPCter or GFP-ImpCter +Lin28-GFPNter) result in significant GFP fluorescence. Quantification of the integrated GFP intensity in different combinations of Imp and Lin28 proteins is shown over hundreds of cells per condition. RFP expression serve as a transfection control and for normalization. Scale bar: 10μm.

significant GFP fluorescence signals be detected, indicating the formation of a stable interaction between the two GFP domains mediated by Imp and Lin28 (Fig 5C). Using these three distinct assays, including in yeast and mammalian cells, we demonstrate that *Drosophila* Imp and Lin28 can interact, and that this interaction is unlikely to be mediated by common mRNA targets, but rather is a direct protein-protein interaction.

## Imp and Lin28 proteins co-localize in vivo

Next, we asked whether Imp and Lin28 proteins may interact in adult intestinal progenitors. Using immunostaining, we observed that Imp and Lin28 endogenous proteins are both expressed in esg-positive cells in similar patterns (Fig 6A). When imaged at higher resolution,

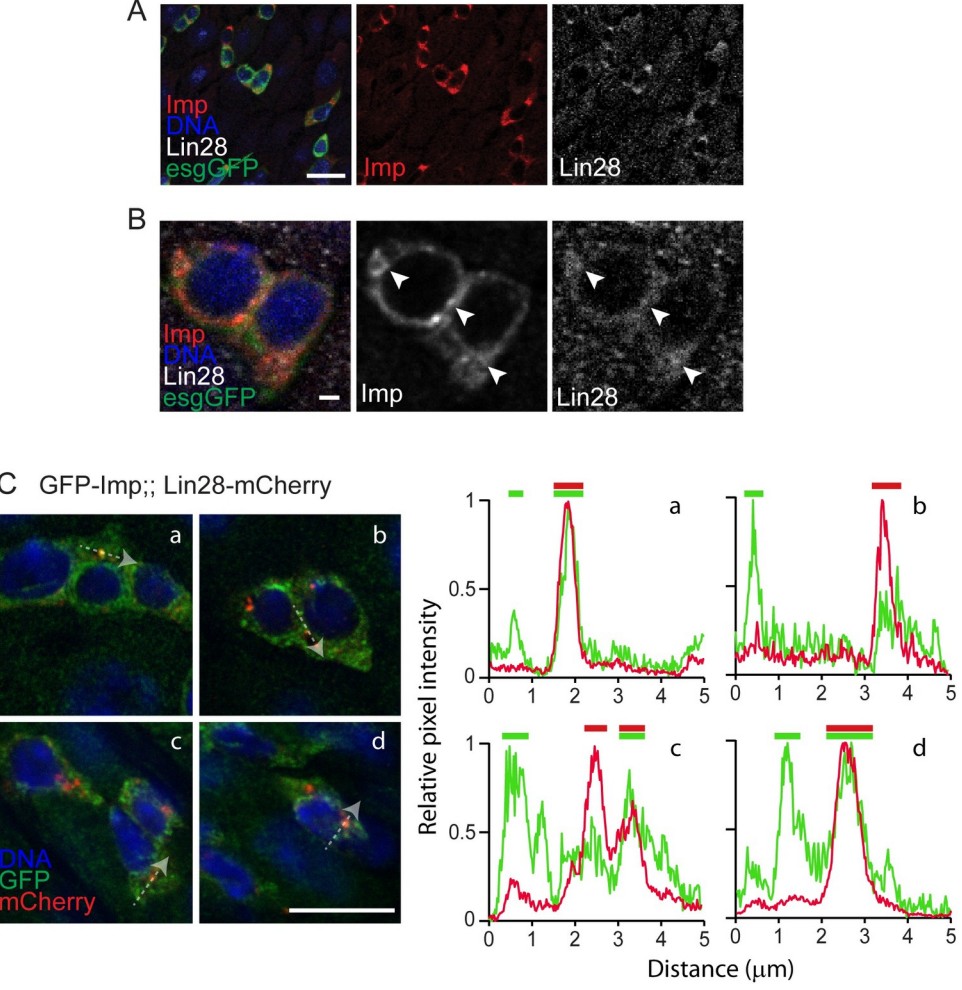

**Fig 6. Imp and Lin28 partially co-localize *in vivo*.** (A) Representative confocal image showing the co-localization of Imp and Lin28 endogenous proteins detected using specific antibodies in wildtype gut. In these 7-day old esgGFP[ts]>GFP kept at 29˚C for 5 days, GFP expression marks esg-positive ISCs and EBs. DNA is stained with Hoechst. Imp and Lin28 proteins are expressed in similar patterns in these cells. Scale bar: 10μm. (B) High magnification images of the same conditions as in (A) show both Imp and Lin28 proteins form detectable and partially overlapping foci in the cytoplasm of esg-positive cells. Scale bar: 1μm. (C) Confocal images of the intestine of GFP-Imp and Lin28-mCherry trans-heterozygotes stained with anti-GFP and anti-mCherry antibodies. Both proteins form partially overlapping cytoplasmic foci. Four groups of esg-positive cells are shown as examples. For each, the relative fluorescence intensity profiles in both GFP and mCherry channels are reported along a 5μm traced line (highlighted by the dashed line arrows). This quantification illustrates the partial correlation between the signals.

both Imp and Lin28 appear cytoplasmic, forming with at least partially overlapping foci (Fig 6B).

Because of the difficulty of imaging endogenous proteins, we next turned to the GFP-Imp and Lin28-Cherry protein fusions to confirm these findings. As described previously Lin28-Cherry appears in few characteristic foci in ISCs and EBs (Fig 6C) [26]. In the same cells, GFP-Imp also forms discernable, although more numerous, foci. Supporting the notion that these foci represent RNP particles, their estimated diameter ranges from 0.5 to 1 micron, a size compatible with several RNPs species [38]. Analysis of both signals indicates that some of these foci contain both GFP and mCherry, while others are either GFP-positive or mCherry-positive, suggesting that Imp and Lin28 partially co-localize in intestinal progenitors (Fig 6C).

We then asked whether stress conditions may affect the sub-cellular distribution of Imp and/or Lin28 proteins. In response to DSS and paraquat, we found that, while GFP-Imp expression is induced (S6A Fig), both GFP-Imp and Lin28-mCherry remain detectable in partially overlapping patterns. Finally, we tested the response to Sodium Arsenite, an ex vivo treatment that robustly induces the formation of Lin28-containing stress granules in ISCs [39]. Like in control conditions or after DSS and paraquat treatments, GFP-Imp and Lin28-mCherry signals remain in partially overlapping cytoplasmic foci (S6B Fig).

These results further suggest that the Imp/Lin28 interaction occurs in adult intestinal progenitors and supports our model in which Imp and Lin28 interact to control the stability of common target mRNAs to regulate ISC proliferation.

## Discussion

In this work, we report that the only Imp or IGF2BP protein present in flies is highly expressed in adult intestinal progenitors and is essential for stem cell proliferation under homeostatic conditions and in response to tissue damage. This significantly improves our understanding of the role of post-transcriptional mechanisms of gene regulation, and more specifically RBPs, in the control of *Drosophila* tissue stem cells. Previous studies have investigated the role of other RBPS, such as Lin28, FMRP and Tis11 [24, 26, 27], our work starts placing Imp in this complex network.

We show that Imp and Lin28 cooperate to cell-autonomously regulate ISC proliferation. Our data demonstrate that when combined, overexpression of Imp and Lin28 leads to synergistic increase in ISC proliferation, while simultaneous loss of function of Imp and Lin28 lead to decreased ISC proliferation, loss of stem cells and short-lived flies. This strongly suggests that Imp and Lin28 act cooperatively, rather than redundantly. In addition, we show that these proteins can form a complex, co-localize in intestinal progenitors, and regulate the expression of a common mRNA target. These observations raise numerous questions regarding the function of the Imp-Lin28 interaction and its impact on stem cell gene expression.

### IMP-Lin28 targets

We found that both Imp and Lin28 control the expression of the InR mRNA in ISCs. Genetically, we showed that InR is a critical target to control ISC proliferation downstream of Imp. In addition, our RNA IP strongly suggest that the InR messenger is bound by Imp and Lin28. Supporting this notion, the large >3kbp 3'UTR of the InR mRNA contains numerous clusters of Imp predicted binding sequences (CAUH, H = A, U, C) and three predicted Lin28 binding sites (GGAGA(U)). Whether Imp and Lin28 can bind the same molecule simultaneously and the effect of Imp and/or Lin28 binding to the InR 3'UTR remains to be investigated. In addition, FMRP has been shown to antagonize Lin28-mediated regulation of InR [26], the effect of Imp and its binding to the InR 3'UTR on this Lin28-FMRP regulation remains to be determined.

Numerous studies in other cell types and organisms demonstrate that both Imp and Lin28 protein families can bind to and regulate the expression of hundreds or thousands of messengers [5, 8, 13]. Thus, it is very unlikely that InR is the only target of Imp in ISCs or EBs in the fly intestine. For example, Imp may regulate the expression of other self-renewal factors whose loss can be compensated for the artificial activation of the insulin signaling pathway. It is also possible that Imp and/or Lin28 may influence cell differentiation in this lineage, as defect in this process would be masked by the block of proliferation resulting from the Imp or Lin28 loss-of-function. While EB-specific Imp knock-down does not affect ISC proliferation, a role of Imp in controlling EB differentiation and cell fate transition in this lineage remains to be specifically tested. Finally, EE differentiation remains poorly understood and the low levels of Imp detected in these cells may be hinting at a role of Imp in these cells. Generally, unbiased approaches will be required to comprehensively identify the mRNA targets of Imp and Lin28 in the different intestinal cell populations and investigate the possible additional functions of these RBPs in adult gut homeostasis and regeneration.

### Imp and Lin28 regulation and interaction

In mammalian stem or cancer cells, growing evidence reveals the aberrant regulation of LIN28s and IMPs expression under pathological conditions. Intriguingly, studies found that LIN28s can directly regulate the expression of IMPs or that, conversely, that IMPs can control the expression of LIN28s. In addition, both proteins are co-deregulated in several cancer cell types, mimicking the synergistic effect that we observed when Imp and Lin28 are simultaneously over-expressed in ISCs. Altogether, this strongly suggests that these two families of proteins form a conserved and complex regulatory network, controlling the expression of critical cell pluripotency and self-renewal factors, across cell populations, tissues, and organisms. In flies, the 3'UTR of Imp contains predicted binding sequences for Lin28, Imp itself or FMRP; the 3'UTR of the Lin28 messenger contains predicted FMRP sites; and the FMRP 3'UTR carries IMP and Lin28 consensus sequences (not shown). We found that over-expressing Imp or Lin28 in esg-positive cells leads to detectable increase in expression of the other factor (S3 Fig), suggesting that Lin28 may directly control Imp expression, while Imp is likely to regulate Lin28 levels indirectly. Altogether, this strongly suggests that, like in mammalian stem cells, auto-regulatory loops control RBPs expression to maintain ISC proliferative homeostasis.

We found that Imp and Lin28 cooperate and directly interact to regulate at least one common mRNA target, but the exact function of this interaction remains elusive. One possibility is that this interaction and the establishment of an Imp-Lin28 network allows ISCs to integrate various external stimuli to adapt their proliferation rate to tissue demand. Supporting this notion, Imp expression was found to be stress responsive, downstream of the ets21c transcription factor and the JNK pathway [25]. On the other hand, Lin28 is controlled by the fly nutritional state [24]. In our hand, we found that Imp protein levels in ISCs and EBs increases after exposure to DSS or paraquat treatment, while the expression Lin28 does not significantly change (S6A Fig). Additional studies will be required to test whether one or both RBPs are regulated downstream of other key signaling pathways controlling ISC proliferation, such as the BMP, EGFR/MAPK, JAK/Stat, Wnt or Hippo pathways.

## Materials and methods

### *Drosophila* Stocks and Culture

The following stocks were obtained from the Bloomington Drosophila Stock Center: $w^{1118}$, *UAS-Lin28*$^{RNAi}$ (50679), *esg-lacZ* (10359), *neoFRT82B*, *neoFRT19A*. *UAS-Imp*$^{RNAi}$ (20322) was

obtained was from Vienna Drosophila Resource Center. MARCM82B and MARCM19A was a gift from N. Perrimon, *UAS-lin28* was from previous study. *FRT19A.Imp7/FM6* and *Imp7/ FM6* were a gift from F. Besse. *UAS-Imp* was a kind gift from P. MacDonald. Intestinal progenitor-specific drivers are esgGal4,UAS-GFP;tub-Gal80ts (termed esgGFP^ts throughout this manuscript), GBE-Su(H)Gal4,UAS-GFP;tubGal80^ts (termed GBE-GFP^ts) and esgGal4,UAS-YFP, tubGal80^ts;Su(H)Gal80 (termed ISC-YFP^ts as Su(H)Gal80 inhibits Gal4 activity in EBs) are from previous studies from B. Ohlstein and S. Hou. The GFP-Imp Trap was generously supplied by L. Cooley; *Lin28-Venus*, *Lin28^Δ1*, *Lin28-Venus* from N. Sokol.

The flies were reared in standard cornmeal/agar medium supplemented with yeast at 25˚C with 60±5% relative humidity and 12h light /dark cycles unless specified.

## Conditional Expression of UAS-Linked transgenes

For temperature sensitive experiments using esgGFP^ts, ISC-YFP^ts and Su(H)GBEGFP^ts, temperature sensitive GAL80ts was used to suppress the early activity of Gal4 before adulthood by rearing them at 18˚C. 2–3 days after eclosion, the adult flies were shifted to 29˚C to activate Gal4 and maintained at 29˚C in standard food unless specified. For control, these gal4 drivers were crossed with $w^{1118}$.

## Mosaic analysis with a repressible cell marker clones

For the mosaic analysis with a repressible cell marker (MARCM) experiments 4–5 days old flies were heat shocked three times at 37˚C for 45 min -1 hour within one day, following which the flies were maintained at 25C. Following MARCM stocks: MARCM82B (hsflp,UAS-GFP; tub-gal4; FRT82B tub-Gal80) and MARCM 19A (hsflp, tub-Gal80 FRT19A; tub-gal4,UAS-mCD8-GFP) were used for inducing somatic clones. Appropriately aged female flies were dissected to analyze the induction of GFP positive clones. At least 6–7 guts were analyzed for each genotype to quantify the number of cells per clones in the posterior midgut region. The different genetic conditions were compared using unpaired two-tailed Student's t-test.

## Stress exposure

Dextran Sodium Sulphate (DSS, 4%; Sigma-Aldrich) and Paraquat (5mM; Sigma-Aldrich) were used for *in vivo* stress experiments. Young adult flies were starved for 4hours followed by feeding them with appropriate stressor in 5% sucrose-saturated filter paper. Intestines were dissected 24 or 48 hours after treatment start as indicated in figure legends.

Sodium Arsenite treatment was performed as describe previously [39]. In short, guts from seven-day-old animals were dissected in Schneider's media and incubated for 1 hour in Schneider's media with or without 10mM Sodium Arsenite. Following treatment, guts were washed and processed for regular immunohistochemistry.

## Immunohistochemistry

Immunostaining was done as previously described. Briefly, appropriately aged guts were dissected in PBS. The dissected guts were immediately fixed in glutamate buffer containing 4% Formaldehyde for 20 minutes followed by a series of methanol washes as previously described [37]. The fixed guts were washed in PBS containing 0.1% Triton X-100 and 0.5% BSA (Gut Buffer). It was then blocked-in gut buffer for one hour followed by overnight incubation at 4˚C in appropriate antibody. Secondary antibodies were used at 1:500. After the secondary antibody treatment, the guts were mounted on slides with Mowiol/Dabco solution. For pH3 staining the guts were fixed for 45 minutes in PBS with 4% formaldehyde.

The following primary antibodies were used: Sox21a antibody was previously generated in the lab. The anti-Delta (C594.9B), anti-Prospero (MR1A), anti-β-galactosidase (40-1a) were obtained from the Developmental Studies Hybridoma Bank (DSHB) and the Anti-phospho-Histone H3 (06–570) from Millipore. Anti-GFP was obtained from Thermofisher Scientific and anti-mCherry was obtained from Biovision. Anti-Imp was a gift from P. Macdonald. Anti-Lin28 was a gift from N. Sokol. Fluorescent secondary antibodies were obtained from Jackson Immunoresearch. Hoechst 33258 (Sigma Aldrich) was used to stain DNA.

## Imaging and signal analysis of the intestinal epithelium

For all quantifications of pH3 positive cell numbers, pH3+ cells were manually counted over the entire intestine using a Zeiss Imager M2M fluorescence microscope. Data are represented as average ± SEM and p values were calculated using an unpaired two-tailed Student's t test.

Confocal images were collected using a Leica SP5 confocal system and processed using the Leica LAS-AF software, analyzed using the Fiji software package and assembled using Adobe Photoshop and Illustrator. All the images were captured from the R4 region of the midgut [40].

For the quantification of signal intensity in S4B Fig, raw images of single confocal z-section were analyzed using the Fiji software package. The outline of esg-positive cells was manually drawn and the average pixel intensity in each region of interest calculated. These values were normalized by subtracting the average pixel intensity in an adjacent background region of comparable size. These normalized pixel intensities are presented.

For the analysis of the Imp and Lin28 localization in Fig 6B, 5μm x 3 pixels lines were drawn over representative single z-section confocal images to overlap with GFP and/or mCherry foci. The "Plot Profile" function of Fiji was used to collect signal intensity values along the lines and traces were generated using Microsoft Excel.

## Gene expression analysis by quantitative RT-PCR

Total RNA was extracted from 6 midguts in triplicates from age-appropriate flies using RNeasy Mini Kit (QIAGEN) according to manufacturer's instruction. cDNA was synthesized after DNAse1 treatment using Superscript III reverse transcriptase (Invitrogen) according to manufacturer's instruction. Real- time PCR was performed on Applied biosystems QuantaStudio 5 using Quantabio Perfecta SYBR green mix according to manufacture protocol using the following primers.

InR Forward 5' GAGGAGAAGCAGCATGGATATAG 3',
InR Reverse 5' CCCTAATTTGCAGGCATAGAGA 3';
rp49 Forward 5' CCAGTCGGATCGATATGCTAAG 3',
rp49 Reverse 5'CCGATGTTGGGCATCAGATA 3',
esg Forward 5' CCGCCCATGAGATCTGAAAT 3',
esg Reverse 5' GGTGATGATGGGTATGGGTATAG 3'.

Relative expression was calculated using the ΔΔCT method and normalized to esg levels in Fig 4B or rp49 levels in Fig 4C. All the qPCR experiments were performed using three independent biological replicates. P values were calculated using unpaired two-tailed student's t test.

## Immunoprecipitation (IP)

S2 Cells transfected with HA tagged Lin28 and FLAG Tagged Imp were lysed in RIPA buffer without SDS. The lysate was incubated with Flag antibody for 4 hours at 4˚C and washed Protein A/G beads were added to the mixed for another 2 hours. Following, the beads were

washed 3 times with lysis buffer. The beads were then boiled in 2X sample buffer and analyzed by western blot using either Lin28 or Imp antibody. For immunoprecipitation with RNase treatment, the lysate was treated with RNase (100μg/ml, Thermo Scientific) for 30 minutes followed by standard IP along with appropriate controls.

## RNA-Immunoprecipitation (RIP)

RNA immunoprecipitation was done as previously described [41] with minor modification. 200 guts were lysed using polysome lysis buffer containing protease inhibitor cocktail (Pierce), RNase inhibitor (Invitrogen) along with DTT. The lysed guts were centrifuged at high speed for 15 minute and frozen for 30 minutes at -80°C. The thawed lysate was subjected to Immunoprecipitation using GFP-agarose beads (Allele Biotechnology) for 4 hours at 4°C and washed 5 times in wash buffer followed by 3 times washing in wash buffer with 1M urea. The washed Agarose beads were subjected to DNAse1 treatment for 10 minutes at 37°C followed by proteinase K(10mg/ml) treatment at 55°C for 30 minutes. The proteinase K treated beads were subjected to Phenol: Chloroform: Isoamyl alcohol extraction to extract RNA. Total RNA was subjected to reverse transcription using reverse transcriptase (SSIII- life technologies). The RT samples were used as template to amplify RNA bound to the flagged proteins by Real time PCR. Real Time PCR was performed on QuantStudio-5 (Applied biosystems) using Quantabio Perfecta SYBR green SuperMix with gene specific primers.

## Protein fragment complement assay

HEK293T cells were transiently transfected with vectors to express the mKG_N and mKG_C fusion proteins. The cells were plated on a 24 well plate and transfected using Effectene transfection reagent (Qiagen) with appropriate combination of plasmids. 2 days after transfection the cells were fixed and imaged using a Celigo Cytometer. The acquired images were then analyzed using ImageJ software (NIH).

## Yeast 2-hybrid

The full length CDS *Drosophila* Imp and Lin28 cloned in pACT2 AD vector (Clontech) to produce Gal4 transcriptional activation domain (TAD) and LexA DNA binding domain-(DBD) to generate TAD-Lin28 and TAD-Imp; and DBD-Lin28 and DBD-Imp respectively [42]). All constructs were verified by DNA sequencing. Various combinations of plasmids expressing TAD and DBD were transformed into the yeast strain YPH500 (MAT *α*, *ade 2*, *his3*, *leu 2*, *lys2*, *trp1*, *ura3*) harboring the pSH18-34 plasmid (lexAop- LacZ reporter) by the standard lithium acetate method. Independent transformants for each combination were patched onto glucose plates containing X-gal for 24–48 hours, followed by a beta-galactosidase liquid assay as described previously [43].

## Longevity analysis and stress sensitivity

Lifespan assay was performed as described [44]. Briefly, temperature sensitive tub-Gal80ts was used to suppress the early activity of esg-Gal4 before adulthood by rearing them at 18°C. 2–3 days after eclosion, mated female flies were shifted to 29°C to activate Gal4 and maintained at 29°C on standard food. Adults were reared at 29°C with 60±5% relative humidity in groups of 23 to 25 females. Thrice per week, flies transferred to fresh food vials and deaths recorded.

For measuring survival after stress exposure, 3 groups of 20 young flies per genetic condition were exposed to toxicant and mortality measured daily.

## Statical analyses and data presentation

All statistics reported in this study were calculated using the GraphPad Prism software and data graphs were prepared using Prism or Microsoft Excel. All raw numeral data are presented in the S1 Data—Excel file.

## Supporting information

**S1 Fig. Imp is required in esg-positive cells for stress-induced ISC proliferation.** (A) Plot depicting the quantification of the number of pH3 positive cells in guts with esg-specific knockdown of Imp, Lin28, or Imp and Lin28 following 48-hours DSS treatment. Single perturbations, as well as double knock-down, result in almost complete block of cell proliferation. (B) Plot depicting the quantification of proliferation as measured by the number of phospho-Histone3 (pH3) positive cells per gut in control vs esg>Imp$^{RNAi}$ after 24-hour exposure to paraquat. In both panels, each data point represents the number of pH3 positive cells per gut. Student's t-test **** p-value = 0.0001.
(JPG)

**S2 Fig. Imp and Lin28 requirement for stress response and complementation.** (A) Survival after exposure to DSS. Curves represent that average and standard deviation of 3 populations of 20 flies per conditions. Note that the same control is presented in both panels and curve are presented in 2 groups for better clarity. Simultaneous over-expression of Imp and Lin28 results in significantly greater sensitivity to DSS, while other manipulation does not affect survival under these conditions. Student's t-test * p-value < 0.05 compared to controls at these time-points. (B) Imp and Lin28 are not genetically redundant in ISCs. Representative MARCM images show that $Imp^7$ and $Lin28^{\Delta 1}$ clones fail to grow, while over-expression of Imp and Lin28 respectively, restores significant clone growth. However, over-expression of Lin28 in $Imp^7$ clones or over-expression of Imp in $Lin28^{\Delta 1}$ clones does not rescue the proliferation blockage. The number of cells per clones in the different genetic backgrounds, 5 and 15 days after heat shock induction (AHS), are shown below. Student's t-test **** p-value = <0.0001; ns = > 0.05. Scale bar: 10μm.
(JPG)

**S3 Fig. Imp or Lin28 expression increases expression of the other factor.** Representative confocal images show that elevated Imp and Lin28 proteins can both be detected in ISC/EBs of esgGFP$^{ts}$>UAS-Imp or esgGFP$^{ts}$>UAS-Lin28 animals, 5 days after transgene induction at 29˚C. GFP (green) marks intestinal progenitors. DNA is stained with Hoechst.
(JPG)

**S4 Fig. Imp and Lin28 promote InR expression.** (A) Representative confocal images showing that over-expressing Imp for 7 days at 29˚C in the ISCs and EBs leads to elevated expression of InR protein. GFP expression represents escargot positive cells (ISCs and EBs), Prospero stains for enteroendocrine cells, and Hoechst for DNA. Scale bar: 10μm. B) Quantification of the expression of InR protein in the esg-positive progenitors using Fiji/ImageJ shows that there is a significant elevated expression of InR in cells over-expressing either Imp or Lin28. Each data point represents individual cells quantified.
(JPG)

**S5 Fig. Imp / Lin28 co-immunoprecipitation in S2 cells.** Western blot showing the reciprocal co-immunoprecipitation of Imp and lin28 using Flag-tagged Imp and HA-tagged Lin28 co-transfected in S2 cells. Inputs show expression of the fusion proteins, and

immunoprecipitation using a non-specific IgG serve as controls.
(JPG)

**S6 Fig. Imp and Lin28 partially co-localize in the intestinal lineage under control conditions and after stress.** (A) Representative confocal images showing the expression of GFP-Imp (green) and Lin28-mCherry (red) under control conditions and in response to DSS or paraquat for 24 hours. In all conditions, the two fusion proteins can be detected in partially overlapping pattern in small diploid intestinal progenitors. DNA is stained with Hoechst. (B) Representative confocal images showing the expression of GFP-Imp (green) and Lin28-m-Cherry (red) after *ex vivo* treatment with Sodium Arsenite for 1 hour. Inserts a-f present higher magnification images of Imp, Lin28 and Imp+lin28 foci. Both under control conditions and after NaAsO$_2$, all three types of granules can be detected in intestinal progenitors. DNA is stained with Hoechst. Scale bar: 3μm.
(JPG)

**S1 Data. All numerical values described in this study are reported in this.xls file.** Dataset are organized for each graphs presented in the main figures and supplementary figures.
(XLSX)

## Acknowledgments

We are very thankful to Drs. Perrimon, Besse, McDonald, Ohlstein, Jasper, Cooley and Sokol for providing us with essential *Drosophila* lines. We are grateful to Evan Bourtis for technical assistance in the Biteau laboratory and the fly community at URMC for fruitful discussions.

## Author Contributions

**Conceptualization:** Perinthottathil Sreejith, Benoît Biteau.

**Formal analysis:** Perinthottathil Sreejith, Benoît Biteau.

**Funding acquisition:** Changsoo Kim, Benoît Biteau.

**Investigation:** Perinthottathil Sreejith, Sumira Malik.

**Supervision:** Changsoo Kim, Benoît Biteau.

**Writing – original draft:** Perinthottathil Sreejith, Benoît Biteau.

**Writing – review & editing:** Perinthottathil Sreejith, Benoît Biteau.

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
