## [Decision Letter · Decision Letter 0]

24 Feb 2022

Dear Dr Biteau,

Thank you very much for submitting your Research Article entitled 'Imp interacts with Lin28 to regulate adult stem cell proliferation in the Drosophila intestine.' to PLOS Genetics.

The manuscript was fully evaluated at the editorial level and by independent peer reviewers. The reviewers appreciated the attention to an important topic but identified some concerns that we ask you address in a revised manuscript

We therefore ask you to modify the manuscript according to the review recommendations. Your revisions should address the specific points made by each reviewer.

[LINK]

Yours sincerely,

Ville Hietakangas

Associate Editor

PLOS Genetics

Gregory P. Copenhaver

Editor-in-Chief

PLOS Genetics

Reviewer's Responses to Questions

**Comments to the Authors:**

Reviewer #1: In the Sreejith et al manuscript the authors set out to explore how Imp and Lin28 might corporate to intestinal stem cell (ISC) proliferation and cell fate. Studies have shown that Lin28 plays a key role in progenitor cells by promoting food-induced symmetric cell divisions through posttranscriptional stabilization of the InR mRNA. Furthermore, previous studies suggest that fly Lin28 and Imp might cooperate to stabilize Upd mRNA in the hub (niche) cells of the testis to promote stem cell renewal. The authors therefore set out to explore whether Imp and Lin28 might cooperate in adult ISCs to promote their divisions. They show that Imp is expressed in progenitor cells and, like Lin28, is required in the ISCs to promote their divisions in both homeostatic conditions and following tissue damage. They further show that although Lin28 and Imp can promote ISC divisions on their own, overexpression of both strongly enhances short-term effects on ISC proliferation. This leads the authors to propose that Lin28 and Imp act in a cooperative manner. Consistent with this, knockdown of Lin28 or Imp in the ISCs slightly reduces lifespan, while knockdown of both strongly reduces lifespan. They then show that Imp, like Lin28, binds the InR mRNA, and that OE of both Lin28 and Imp increases InR mRNA levels much more than OE of them on their own, suggesting that they cooperate to stabilize InR levels. Finally, they use different approaches to show that Lin28 and Imp interact directly in vitro and in vivo. While their in vitro data are very solid and suggest that Lin28 and Imp interact directly, the in vivo data are less convincing. All in all, this reviewer find that the majority of the data presented are very solid and that story is of interest to the audience of plus genetics when the following additions:

Major points:

1) Imp expression is supposed to be stress-responsive, suggesting that it might integrate environmental stimuli related to tissue damage. Does loss of Imp reduce survival following DSS- or paraquat-induced tissue damage?

2) In figure 4C the authors show that ubiquitous knockdown of either Imp or Lin28 reduces InR levels in the gut. Does this take into account that the number of ISCs are reduced in these conditions (after 7 days)? It would probably be more accurate to measure InR protein levels in ISCs by immunostaining in this condition – if feasible?

3) The authors show that Lin28 and Imp localize in foci that are partially overlapping (Fig. 6B). This reviewer wonders if there is an increase in the overlap between Lin28 and Imp positive loci in conditions where symmetric ISC divisions are favored (either during regenerative growth or in the context of starvation-refeeding)?

4) This reviewer is not convinced about the Co-IP presented in figure 6C. It is not clear which of the bands correspond to Imp-GFP or Lin28. It would be better to improve it or remove it.

Minor point:

line 208: I believe that the figure reference should be 3A?

Reviewer #2: This manuscript describes an association between Imp and Lin-28 that control intestinal stem cell regeneration. The results are novel, well described and of interest to the broader field of stem cell biology.

I have a few comments that I would like the authors to address:

Lines 137-138 state “Imp7 137 homozygous mutant clones remained constantly small, mostly single Dl-positive Sox21a-positive ISCs”. It is very difficult to observe Dl+ Sox21a+ cells in the Imp7 mutant panel. Perhaps separation of channels or an insert at higher magnification would assist the reader.

Lines 155-156 state “This supports the notion that it acts downstream of the stress-responsive transcription factor ets21c in these cells”. What data allow the authors to make this statement? Is there already a genetic association between Imp and ets21c?

Could an example of the PH3 staining please be shown alongside the graph in Fig 2C.

Lines 178-180 state “However, no significant rescue was observed when Lin28 was expressed in Imp null mutant clones or, conversely, when Imp was expressed in Lin28 null clones (Fig3B)”

The data are in fact significant, so perhaps reword to “very slight rescue”.

Could the authors please explain why in Fig 4A the effect of UAS-InR is the same as UAS-InR[ACT]. Is InR maximally stimulated in the midgut?

I found the relationship between InR mRNA and protein somewhat confusing. Upregulated mRNA was only observed when both Imp and Lin28 were expressed yet upregulated InR protein was observed with either UAS-Imp or UAS-Lin28. Can the authors please offer an explanation.

The Co-IP of Lin28 with IMP is convincing in Fig 5B but the anti-Imp western looks like a smear. Is the protein easily degraded? The same is seen in Fig 6C. Could a comment please be made about these westerns and the expected sizes of the protein bands.

Minor comments:

Line 139 – I think this should refer to Fig 2A, not Fig (2B and 2C)

Line 150 – Please define the ISC-Gal4. Is this Delta-Gal4?

Reviewer #3: Post-transcriptional regulation has emerged as a main regulatory level involved in the control of stem cell proliferation. Among the RNA binding proteins found to play a conserved role in this process are Lin-28 and Imp/IGF2BP. These two proteins have been shown to cooperate for the control of stem cell proliferation in different contexts, in both mammalian and fly systems.

In this manuscript, the authors aimed at investigating if Drosophila Imp is required for the control of Intestinal Stem Cell (ISC) proliferation. As Lin-28 was already known to control this process through the regulation of InR mRNA, which encodes the Insulin receptor, they investigated whether Lin-28 and Imp could cooperate to regulate InR expression.

Combining functional studies and biochemical binding assays, they convincingly show that Lin-28 and Imp cooperate to promote the proliferation of ISCs in vivo and that they interact directly. Furthermore, they show that Imp, similarly to Lin-28, binds to InR mRNA and promotes ISC proliferation by promoting the expression of InR.

That Lin-28 and Imp cooperate for the control of stem cell proliferation is not really novel, but the authors provide an additional strong example that will likely be of interest both for the community of researchers working on intestinal homeostasis as well as for RNA biologists. Although the resolution of the provided Figures was limited, the presented data appear to be of high quality. Furthermore, the authors have combined various complementary approaches to support their conclusions.

Beyond minor points listed below, my major point relates to the lack of information related to the regulation of Imp and Lin-28 expression, which would nicely complement the functional and biochemical assays and help understand the regulatory process under study.

Major point:

The authors describe the expression of Imp and Lin-28 in wild-type contexts, but did not investigate 1) if the two proteins are regulating each other, and 2) if their expression is induced in response to stress.

Addressing point 1 would help interpret the genetic interactions observed in vivo and can easily be done by checking the expression of Imp in Lin-28 mutant and overexpression contexts (and vice versa). Addressing point 2 would help understand whether chemical stress (exposure to DSS) induces proliferation by promoting the expression of Imp and Lin-28.

Minor points:

1) The authors should better describe the different populations of cells found in the intestine (EE is never defined and EB is defined only after the first time it is used).

2) why the authors wonder if Lin-28 and Imp act redundantly (line 175) is not clear, as mutating either of them produces a very strong phenotype. Why they thought expressing Lin-28 would rescue the imp phenotype (and vice-versa) is also not clear. Replacing Figure 3B by Supplementary Figure S2B would better illustrate functional cooperativity.

3) The authors refer to the Imp protein-trap line as Imp-GFP. As the GFP is fused in frame N-terminally, it should rather be termed GFP-Imp (both in the text and in Figures).

4) Figure 4D: the authors should better explained how normalization of qPCR data has been performed.

5) Figure 6B: the arrows in a-d are not positioned correctly (the don’t point to granules under study)

6) line 285: the IP-RT-QPCR performed is NOT indicative of direct binding to InR mRNA.

7) The authors should follow the nomenclature used in flies, ie use capitals for proteins only and italics for transcripts, genes and alleles.

8) a number of typos are left in the text

9) Indicating the nature of the statistical tests used in the Figure legends would be helpful.

**Have all data underlying the figures and results presented in the manuscript been provided?**

Reviewer #1: Yes

Reviewer #2: Yes

Reviewer #3: Yes

PLOS authors have the option to publish the peer review history of their article (what does this mean?). If published, this will include your full peer review and any attached files.

Reviewer #1: No

Reviewer #2: No

Reviewer #3: No

---

## [Decision Letter · Decision Letter 1]

3 Aug 2022

Dear Dr Biteau,

Thank you very much for submitting your Research Article entitled 'Imp interacts with Lin28 to regulate adult stem cell proliferation in the Drosophila intestine.' to PLOS Genetics.

The manuscript was fully evaluated at the editorial level and by independent peer reviewers. The reviewers appreciated the attention to an important topic but identified some minor concerns that we ask you address in a revised manuscript. The reviewer comments can be addressed in writing, no further experiments are necessary.

We therefore ask you to modify the manuscript according to the review recommendations. Your revisions should address the specific points made by each reviewer.

[LINK]

Yours sincerely,

Ville Hietakangas

Academic Editor

PLOS Genetics

Gregory P. Copenhaver

Editor-in-Chief

PLOS Genetics

Reviewer's Responses to Questions

**Comments to the Authors:**

Reviewer #1: Commments uploaded as attachment

Reviewer #2: The authors have now satisfied my concerns.

Reviewer #3: The authors have significantly strenghten their manuscript through the revision process and have addressed all my concerns.

Two minor text changes are still required :

- line 256 : reference should be made to Fig S5, not S4

- line 268 : the title of the paragraph lacks a verb

**Have all data underlying the figures and results presented in the manuscript been provided?**

Reviewer #1: Yes

Reviewer #2: Yes

Reviewer #3: Yes

PLOS authors have the option to publish the peer review history of their article (what does this mean?). If published, this will include your full peer review and any attached files.

Reviewer #1: No

Reviewer #2: No

Reviewer #3: No

---

## [Editor Report · Decision Letter 2]

18 Aug 2022

Dear Dr Biteau,

We are pleased to inform you that your manuscript entitled "Imp interacts with Lin28 to regulate adult stem cell proliferation in the Drosophila intestine." has been editorially accepted for publication in PLOS Genetics. Congratulations!

Yours sincerely,

Ville Hietakangas

Academic Editor

PLOS Genetics

Gregory P. Copenhaver

Editor-in-Chief

PLOS Genetics

Comments from the reviewers (if applicable):

**Data Deposition**

http://datadryad.org/submit?journalID=pgenetics&manu=PGENETICS-D-22-00046R2

**Press Queries**

---

## [Editor Report · Acceptance letter]

1 Sep 2022

PGENETICS-D-22-00046R2 

Imp interacts with Lin28 to regulate adult stem cell proliferation in the Drosophila intestine. 

Dear Dr Biteau, 

We are pleased to inform you that your manuscript entitled "Imp interacts with Lin28 to regulate adult stem cell proliferation in the Drosophila intestine." has been formally accepted for publication in PLOS Genetics! Your manuscript is now with our production department and you will be notified of the publication date in due course.

With kind regards,

Zsuzsanna Gémesi

PLOS Genetics

On behalf of:
